# The Impact of Intensive Care Unit Nurses’ Burnout Levels on Turnover Intention and the Mediating Role of Psychological Resilience

**DOI:** 10.3390/bs14090782

**Published:** 2024-09-06

**Authors:** Ömer Turunç, Abdullah Çalışkan, İrfan Akkoç, Özlem Köroğlu, Güney Gürsel, Ayhan Demirci, Köksal Hazır, Neslihan Özcanarslan

**Affiliations:** 1Business Administration Department, Faculty of Economic, Administrative and Social Sciences, Antalya Bilim University, Antalya 07110, Turkey; omer.turunc@antalya.edu.tr; 2Health Management Department, School of Health Sciences, Toros University, Mersin 33140, Turkey; abdullah.caliskan@toros.edu.tr (A.Ç.); ozlem.koroglu@toros.edu.tr (Ö.K.); 3Faculty of Health Sciences, Izmir Tınaztepe University, İzmir 35400, Turkey; dr.irfanakkoc@gmail.com; 4Department of Software Engineering, Faculty of Engineering and Architecture, Konya Food and Agriculture University, Konya 42080, Turkey; 5Department of International Trade and Logistics, Faculty of Economics Administrative and Social Sciences, Toros University, Mersin 33140, Turkey; ayhan.demirci@toros.edu.tr (A.D.); koksal.hazir@toros.edu.tr (K.H.); 6Department of Nursing, Faculty of Health Sciences, Toros University, Mersin 33140, Turkey; neslihan.ozcanarslan@toros.edu.tr

**Keywords:** burnout, turnover intention, psychological resilience, intensive care, nurses

## Abstract

Background: This study aims to investigate the relationship between burnout levels among intensive care unit (ICU) nurses, turnover intention, and the mediating role of psychological resilience. Methods: This survey design was used to collect data from 228 ICU nurses from hospitals in Turkey. The study utilized self-report questionnaires to measure burnout levels, turnover intention, and psychological resilience. The data were analyzed through structural equation modeling. Results: In the study, a positive relationship between burnout and turnover intention and burnout and psychological resilience is significant (*p* ≤ 0.001). However, psychological resilience was not found to have a mediating role, indicating that other factors or variables may play a more substantial role in the relation found between burnout and turnover intention. Also, the research model’s Standardized Root Mean Square Residual is expected to be less than 0.10 for the model to have an acceptable fit. Conclusion: The findings suggest that levels of burnout among ICU nurses may have significant consequences on turnover intention. However, promoting resilience cannot help diminish the negative effects of burnout on turnover intention. The findings highlight the importance of burnout levels in nursing by synthesizing findings from the previous literature. Then, focusing on the concepts of turnover intention and psychological resilience, it explains the importance of these concepts in the Intensive Care Unit nurses and their relationships with each other.

## 1. Introduction

Burnout, the one of important dimensions of stress in healthcare, is encountered more frequently due to work intensity, low wages, hopelessness, the number of patients, and unfavorable working conditions. Burnout, which is the precursor of many variables, has a significant interaction, especially to leave the job [1]. The turnover is particularly critical, given its significant repercussions on the overall organizational outcomes, including the consistency of patient care, potentially leading to heightened adverse events and patient mortality [2,3].

The decision-making process of nurses, whether to stay in their current position or to leave, is influenced by a myriad of factors. These range from personal factors such as age, education, gender, marital status, and work experience [4,5,6], to organizational factors like team support, organizational support, work autonomy, adequate staffing, a positive work climate, and effective role modeling by leaders [7]. Among these, psychological stress and emotional exhaustion stand out as significant predictors of nurses’ intentions to leave their organizations [4,7,8].

Several stressors contributing to nurse burnout are identified, including lack of occupational competency, complex patient circumstances, insufficient resources, and inadequate workplace procedures [4,9]. The direct link between burnout and turnover is notable, particularly among intensive care nurses, who face elevated levels of burnout and demonstrate a strong inclination to resign from their organizations [10,11]. This burnout is also linked to lower assessments of patient care quality and increased absenteeism and turnover intentions [12,13]. 

In the context of Turkey, nursing emerges as a challenging profession due to the exceedingly low nurse-to-population ratio (30.03 per 10,000), ranking it 93rd among 193 countries (WHO, 2021), coupled with difficult working environments [14]. Nurses, bearing the brunt of global health crises, often experience stress and excessive workload, leading to burnout [15,16,17]. The lack of clearly defined performance criteria and job expectations further exacerbates this situation [18]. Nurses, especially those in intensive care, are considerably affected by their work environment, often leading to mental health problems which in turn can detrimentally affect patient care quality [13,19,20,21]. 

The study suggests that the negative factors affecting nurses do not solely amplify their level of burnout and intent to quit the profession; psychological resilience also plays a significant role. Defined as a personality trait enabling individuals to recover from extreme trauma, tragedy, threat, or stressors [22,23,24], psychological resilience is demonstrated to be associated with nurse burnout and turnover intentions [25,26]. 

This study aims to examine the role of psychological resilience in the relationship between nurses’ burnout levels and turnover intention. Psychological resilience, which can be defined as a person’s ability to successfully overcome these negative conditions, adapt, and be resilient despite very difficult conditions, is an important factor for the nursing profession. This exploration aims to understand how psychological resilience can influence the relationship between the burnout experienced by nurses and their subsequent decisions regarding continuing or leaving their profession. Although there are many similar studies in the field, the fact that the study was conducted in a sample of intensive care nurses and that the mediating role of psychological resilience, a relatively new and important variable, was investigated differentiates the study from similar studies.

Turnover intention, indicative of an employee’s desire to leave their organization within a specific timeframe, is a key precursor to actual job separation [27]. Studies reveal varying nurse turnover rates across different regions, influenced by both individual factors (salary, experience, job title) and organizational factors (workplace stress, family support, job satisfaction, burnout, organizational support, and commitment) [28]. Nurse turnover is not only costly due to temporal displacement but also leads to increased physical and emotional distress for patients [29]. Research across twelve European countries in the RN4CAST project has shown that higher nurse turnover intentions correlate with lower patient satisfaction in hospital care, further impacted by burnout and job dissatisfaction [2]. 

The adverse effects of nurse turnover extend to the nurses themselves, the patients, and the broader organization [30,31]. This is particularly evident in specialized areas such as ICUs, where the departure of experienced nurses adversely affects nursing services, increasing error rates and patient mortality [32]. In human resource management, employee turnover is a persistent issue, incurring significant costs associated with turnover intentions [33]. Burnout, a major cause of turnover intention, is positively correlated and may act as a mediator between the nursing practice environment and turnover intention [34,35,36,37]. 

Burnout, defined as prolonged occupational stress leading to emotional exhaustion, cynicism, and reduced personal achievement [38], is a notable factor in nursing. In nursing, burnout is encountered more frequently due to work intensity, low wages, hopelessness, the number of patients, and unfavorable working conditions. These conditions can lead to various negative outcomes, including high turnover levels [39,40]. Several studies affirm that burnout levels significantly influence turnover intentions [41,42], defining burnout as a reaction to ongoing stress, encompassing emotional exhaustion, physical fatigue, and cognitive impairment [43]. 

The study specifically targets ICU nurses who consistently confront high quantitative work demands, such as work overload, identified as a significant stressor and predictor of burnout and turnover, especially when job resources are limited [44]. Consequently, the study proposes the following hypothesis:

**Hypothesis 1**.*Burnout levels in intensive care unit nurses positively and significantly affects their turnover intention*.

The role of psychological resilience in nursing is also a focal point of this study. Recognized as a crucial factor in reducing burnout and turnover, psychological resilience provides essential support in managing the challenges of demanding job requirements [45]. This ability to effectively manage and overcome difficulties is especially relevant in high-change and stress scenarios, such as those faced in nursing [46]. The relationship between psychological resilience, coping with stress, and conversion disorder has been extensively studied, highlighting the necessity of resilience in nursing to ensure effective and safe patient care [47,48,49,50,51]. Psychological resilience is crucial for practicing nurses, aiding them in coping with stress and preparing them for professional roles [52,53,54]. 

Resilience enables individuals to succeed despite adversity, influenced by factors like impulse control, optimism, self-efficacy, problem analysis, emotion regulation, seeking support, and empathy [55,56]. The study therefore proposes the second hypothesis:

**Hypothesis 2**.
*Burnout levels in intensive care unit nurses negatively and significantly affect their psychological resilience.*


Individuals with high levels of resilience are better equipped to maintain optimal performance and energy levels, even under increased stress. Lack of sufficient resilience can lead to negative outcomes such as decreased performance, job satisfaction, commitment, and motivation [57]. Research indicates that resilience is positively associated with job satisfaction and negatively associated with quitting [58,59]. This underscores the importance of developing resilience-building strategies as part of human resource management practices to enhance employee well-being and retention. The study therefore proposes the third hypothesis:

**Hypothesis 3**.
*The level of psychological resilience in intensive care unit nurses negatively and significantly affects their turnover intention.*


Resilience is also significant in various professional domains, including nursing and physical education. It enables individuals to employ active coping strategies, effectively navigate and overcome workplace challenges, and reduce emotional exhaustion and turnover intention [60,61,62,63]. The link between high resilience and low rates of burnout and job separation is well documented in nursing, with significant correlations observed between nurses’ well-being, resilience, and turnover [64]. Resilience has been shown to mediate the relationship between burnout and psychological distress, suggesting its potential role as a mediator in the relationship between emotional labor, burnout, and turnover intention in nursing [64,65,66]. The study therefore proposes the fourth hypothesis:

**Hypothesis 4**.
*Psychological resilience mediates the relationship between burnout and turnover intention in intensive care nurses.*


## 2. Materials and Methods

### 2.1. Study Design

This cross-sectional study analyzed the moderating effect of psychological resilience on the relationship between burnout and turnover intention in intensive care unit nurses. The research model and hypotheses are presented in Figure 1.

### 2.2. Sampling and Participants

The population of the study (*n* = 450) consisted of ICU nurses working in the hospitals of a university operating in Antalya, Turkey. A total of 300 questionnaires were distributed but 245 were collected (81%). After excluding the missing data (respondents who left the questionnaires blank or significantly incomplete), 228 questionnaires were found suitable for analysis. Demographic factors of the nurses who participated in the study were as follows: 51% were female (*n* = 116), 46% (*n* = 104) were married, and 25% (*n* = 57) graduated from a university. The mean age of the sample was 29 years (SD = 7.6), and the mean years of employment was 6.6 years (SD = 5.2).

### 2.3. Instruments

The research questionnaire was distributed and collected as a paper questionnaire to intensive care nurses. Due to the workload of the nurses, a significant portion of the questionnaires were conducted face-to-face to ensure that the answers were objective.

The psychological resilience scale developed by Connor and Davidson [67] was utilized in this study to measure the psychological resilience level of ICU nurses. The scale comprises 25 items adding 3 dimensions. For this research, 25 items of the psychological resilience level were employed. Karaırmak [68] conducted the Turkish validation of the scale, and the reliability analysis appeared to have a Cronbach’s alpha reliability coefficient of 0.92. Although this scale is 5-dimensional, as a result of the analysis, the 5-factor structure of the scale could not be confirmed in our sample, and it was included in the analysis with its single-factor structure. The PR scale has also been adapted for some cultures and used as a single factor [68,69,70].

The Maslach Burnout Scale was developed by Maslach and Jackson [38] and adapted into Turkish by Ergin [71] to assess burnout. The scale consists of 22 items that are designed to measure three dimensions of burnout: emotional exhaustion (9) depersonalization (5 items) and personal accomplishment (8 items). Ergin [71] reported Cronbach’s alpha reliability coefficients for the scale, respectively, 0.83, 0.71, and 0.72. The Cronbach Alpha coefficient calculated separately for each sub-dimension of the scale was 0.84 for emotional exhaustion, 0.80 for depersonalization, and 0.76 for personal accomplishment. Although the burnout scale is 3-dimensional, as a result of the analysis, the 3-factor structure of the scale could not be confirmed in our sample, and it was included in the analysis with its single-factor structure.

As a matter of fact, factor analyses are conducted as scale-dependent, which may differ according to the sample and culture, and the factor structure compatible with the sample is used [72]. Factor analysis is an important statistical operation in the social sciences since it elucidates the quality and validity of measurement. The primary objective of factor analysis is to reduce the number of dimensions [73].

In the meta-analysis of Worley et al. [74], one of the studies conducted to examine the structural status of the MBI, it was reported that although the 3-factor structure of the MBI is generally preserved, its dimensions can vary between 1 and 4 factors. Walkey and Green [75], Densten [76], Schneider et al. [77], and Galanakis et al. [78] analyzed the MBI with less than three dimensions.

According to some researchers [79,80], the so-called existential model of burnout [81] and the phase model proposed by Golembiewski [82] are considered one-dimensional as they represent burnout as a single state [83].

The study used a six-item Rusbult et al. [84] and Wayne et al. [85] scale to measure turnover intention and adapted to Turkish by Erdirençelebi and Ertürk [86]. Erdirençelebi and Ertürk [86] reported a Cronbach’s alpha reliability coefficient of 0.88 for the scale.

Answers in the scales were taken with a 5-point Likert scale (1 = strongly disagree, 5 = strongly agree).

All scales used in the study were adapted from the original scales, validity and reliability studies were conducted, and they were taken and used regarding studies published in open sources.

### 2.4. Data Collection

These cross-sectional data were gathered by self-report online survey in Antalya, Turkey, from April to May 2023. Respondents were nurses working in public hospitals. Questionnaires were applied to 300 nurses by using the convenience sampling method and 228 eligible questionnaires were utilized for data analysis, with an 81% response rate. Nurses participated in this study voluntarily and all participating nurses have informed verbal consent before completing the questionnaire.

### 2.5. Ethical Considerations

The research was conducted in compliance with ethical standards, having received approval from the Antalya Bilim University Ethics Committee (Decision No: E-12402273-640.640.640 300003995). Informed consent was provided by all participants, and necessary permissions were obtained from the institution where the research was conducted. All the participants in this study were informed about the purpose of the study before being asked to fill out the questionnaire. In addition, autonomy to participate in the study was guaranteed, and all information was kept confidential and used only for scientific research. Anonymity was assured by using anonymous surveys that cannot be traced back to the respondent. The survey contained no personally identifiable information such as name or contact information. All responses gathered were combined and summarized in the report to further protect participants’ anonymity.

### 2.6. Statistical Analysis

Data analysis was performed using SPSS (version 22) and the significance level was determined at a 95% confidence interval. At the very first stage, the appropriateness of the measurement model to the data was checked.

To reveal the structural validity of the research model, a confirmatory factor analysis was conducted. In this analysis, the maximum likelihood estimation method was utilized. Scales were analyzed for reliability using Cronbach’s alpha reliability coefficient. The descriptive analyses were computed to define the sociodemographic characteristics. Pearson correlation analyses were used to examine the relationship among the study variables. Following the preliminary analyses, the mediation model was analyzed using AMOS and Smart PLS. The mediation model was tested by using structural equation modeling and bootstrap method.

## 3. Results and Discussion

The measurement model was first tested to examine the Discriminant Validity Fornell–Larcker Criterion (Table 1) construct reliability and validity (Table 2) for the study’s variables. The measurement model consists of the study variables that contain scale items reflecting the relevant implicit constructs. Three variables comprise the measurement model: turnover intention (six statements), burnout (twenty-two statements), and psychological resilience (twenty-five statements).

Table 3 presents the findings of reliability, average, standard deviation, and correlation analysis of the data collected from nurses related to turnover intention, burnout, and psychological resilience. The research model found some significant relationships among the dependent, independent, and mediating variables (Table 3). Correlation analysis revealed that there were positive relationships between the independent variable (burnout) and the dependent variable (turnover intention), and between the mediating variable, and the dependent variable (psychological resilience–burnout). On the other hand, there is no meaningful correlation between the mediating variable (psychological resilience), and the dependent variable (turnover intention).

At this research stage, mediation effects were tested with the bootstrap method proposed by Hayes [87]. The bootstrap method with 5000 resampling [87] was used to test the effects of path analysis results (total, indirect, mediation, and moderation) among the variables in this study model (Table 4).

The indirect effect of psychological resilience, the mediating variable in this study, has been explained using the bootstrap method. The mediating role of the mediating variable in the relationship between the independent and dependent variables is not significant when both bootstrap values are measured together at a 95% confidence interval [88]. As a result of the analysis of the independent variable in the research model on the dependent variables, path coefficients (Table 4), and total and indirect effects are shown in Table 4.

According to the path analysis that shows the path coefficients of the relations between the variables in the model (Table 4); the coefficient of the paths between burnout and turnover intention and burnout and psychological resilience are significant (*p* ≤ 0.001). According to these findings, Hypothesis 1 is supported, and this finding is consistent with similar studies as expected [41,42,58]. Burnout is an appropriate factor to predict turnover intention and is also supported by meta-analytic studies [89,90,91,92]. In nursing, the prevalence of burnout is typically associated with highly demanding and intense working conditions and can lead to a variety of negative outcomes, including high levels of turnover [39]. Numerous studies have demonstrated that burnout levels affect turnover intentions [41,42,58].

According to these findings, Hypothesis 2 is not supported. This finding is unexpectedly not in line with previous research. Indeed, studies in the literature show divergence from this finding. Research by Hudgins [58] showed that higher levels of resilience are positively related to increased job satisfaction among employees, while resilience is correlated negatively with turnover intention.

As seen in Table 4, the mediation effect is considered significant according to indirect effects [42]. It is concluded that there is no indirect relation between burnout, psychological resilience, and turnover intention.

Based on this finding, Mediation Hypothesis 3 is not supported [89]. This finding is unexpectedly not in line with previous research. Indeed, studies in the literature show divergence from this finding. Wibowo and Paramita’s research shows no significant relationship exists between psychological reliance and turnover intention in nurses [93].

The findings regarding some of the criteria required to determine the fit of the research model within the scope of the Smart PLS analysis methodology are presented in Table 5. SRMR (standardized root mean square residual) value is expected to be less than 0.10 for the model to have an acceptable fit. In the present study, the SRMR value for the model was calculated as 0.010. The d_ULS (the squared Euclidean distance) and d_G (the geodesic distance) values calculated as 1.072 and 0.925, respectively, did not meet the perfect fit criteria of *p* > 0.05. The chi-square value (a statistical test used to examine the differences between categorical variables from a random sample to judge the goodness of fit between expected and observed results) was determined as 493.690. The NFI (normed fit index) value is expected to be in the range between 0 and 1 and a value close to 1 indicates a good fit for the model. In this study, the NFI value was calculated as 0.08 for the model.

As a result of the analyses conducted in the same direction as the hypotheses of the study, the relationships of the three dimensions of the burnout scale (emotional exhaustion, personal accomplishment (R), and depersonalization) with the dependent variable and the possible mediator/moderator variable could not be reported due to the construct validity problem. Although it is stated before that construct validity and reliability of the unidimensional burnout scale was confirmed by the literature [72,73,74,75,76,77,78,79,80,81,82,83], it is considered that eliminating this deficiency by addressing the three dimensions of burnout separately and with a different methodology could significantly contribute to the nursing research.

For this purpose, hierarchical regression analyses were conducted by SPSS 22, the burnout scale dimensions were considered separate scales, and their effects on the independent variable turnover intention and the moderating role of psychological resilience in this interaction were examined. By these analyses, the independent variable burnout dimensions and the moderating variable psychological resilience were centralized, and the variables of age and tenure were controlled.

According to these analyses, it was determined that emotional exhaustion had a positive and significant effect on turnover intention (β = 0.46; *p* ≤ 0.001), but psychological resilience did not play a moderating role in this relationship (β = −0.01; *p* > 0.05). In addition, it was determined that personal accomplishment (R) also positively and significantly affected turnover intention (β = 0.28; *p* ≤ 0.001), but psychological resilience did not play a moderating role in this relationship (β = −0.06; *p* > 0.05). In the last stage, it was determined that depersonalization also positively and significantly affected turnover intention (β = 0.37; *p* ≤ 0.001), but psychological resilience did not play a moderating role in this relationship (β = −0.07; *p* > 0.05).

## 4. Conclusions

The present cross-sectional study aimed to examine the mediating effect of psychological resilience on the relationships between burnout and turnover intention among ICU nurses. The findings revealed several important insights into the relationships between these variables.

First, the results showed a positive relationship between burnout and turnover intention among ICU nurses. Thus, this finding suggests that greater levels of burnout are more associated with turnover intention. It highlights the importance of addressing burnout as an essential factor in reducing turnover intention among nurses in the intensive care unit. This finding is consistent with similar studies as expected. Burnout is an important factor for predicting turnover intention and is supported by meta-analytic studies. Numerous studies have shown that burnout level affects turnover intention. Based on this finding, it is reiterated that hospital and health department managers should take measures to reduce the inputs that may cause burnout to ensure that intensive care nurses, who have a very important role in the patient treatment process, stay at work and reduce their turnover rates. This finding tested the validity of an expected and known effect on intensive care nurses who are qualified labor force.

Secondly, a significant positive relationship between burnout and psychological resilience was found in the study. This indicates that as burnout levels increase, psychological resilience increases. It suggests that burnout may have a supporting effect on the psychological resilience of ICU nurses. This finding is unexpectedly not in line with previous research. Indeed, studies in the literature show divergence from this finding. Research showed that higher levels of resilience are positively related to increased job satisfaction among employees, while resilience is negatively related to turnover intention.

It is believed that the reasons for this discrepancy may be attributed to the unique conditions of intensive care nurses and the different outcomes of their intense and taxing experiences. It may be challenging to positively regulate the factors leading to nurses’ attrition. The findings of the studies have also demonstrated that even psychological resilience may not influence the decision to leave the job. It is considered that employment in this field, which is related to human health, cannot be easily abandoned for simple reasons.

Third, psychological resilience did not have a significant mediating role in the relationship between burnout, and turnover intention. This finding suggests that psychological resilience does not indirectly affect the relationship between burnout and turnover intention in ICU nurses. Other factors such as age, gender, tenure, or variables may play a more important role in the relationship between burnout and turnover intention. Although it was found that psychological resilience does not play a role in the relationship between burnout and turnover intention, it is thought that it is necessary to state that it would not be surprising to encounter unexpected results in professions with special characteristics, especially in jobs such as intensive care nursing, which require intense sacrifice related to human health. The situations and workload faced by intensive care nurses already necessitate resilience [58].

Overall, these findings highlight the importance of addressing burnout and turnover intention among ICU nurses. Interventions should focus on reducing burnout levels and promoting positive factors like workload management, social support, self-care practices, and resilience-building programs. Such interventions may be beneficial in mitigating burnout and reducing turnover intention. In contrast to expectations, some positive variables such as psychological resilience may not demonstrate the desired effects on the work climate of intensive care nurses. The findings of this study suggest the need to examine different attitudes and behaviors to promote the expected behaviors in the unique and intense work environment of intensive care nurses, which may diverge from theoretical predictions

It is important to note that this study has some limitations. The research design was cross-sectional, which limits causal inferences. Longitudinal studies may provide more robust evidence on the associations among burnout, resilience, and turnover intention. Moreover, this study was undertaken in a particular geographical region, which might limit the generalizability of the findings. Future research should consider larger and more diverse populations to increase the external reliability of the results.

Despite the confirmation of construct validity and reliability of the unidimensional burnout scale by the literature, the three dimensions of burnout are separately examined with a different methodology and added to the findings. With this approach, an important limitation of the research was eliminated.

Overall, this study sheds light on the complex relationships between burnout, psychological resilience, and turnover intention among ICU nurses. It emphasizes the need for organizational interventions and support systems to address burnout and promote psychological well-being in the healthcare workforce. By implementing strategies to reduce burnout, and enhance psychological resilience, healthcare organizations can contribute to improving nurse retention and ultimately provide better patient care in the ICU setting.

In the literature, there are many studies on burnout and turnover intention in a sample of nurses. The most important differences distinguishing this study from similar studies are the study was conducted in a sample of intensive care nurses, it investigated the mediating role of psychological resilience, which is a relatively new and important variable. The findings of the study that do not comply with the literature is another significant point.

Although psychological resilience was not found to play a role in the relationship between burnout and turnover intention, future research may examine the mediating or moderating roles of other factors such as age, gender, tenure, or variables to determine their role in the relationship between burnout and turnover intention.

## Figures and Tables

**Figure 1 behavsci-14-00782-f001:**
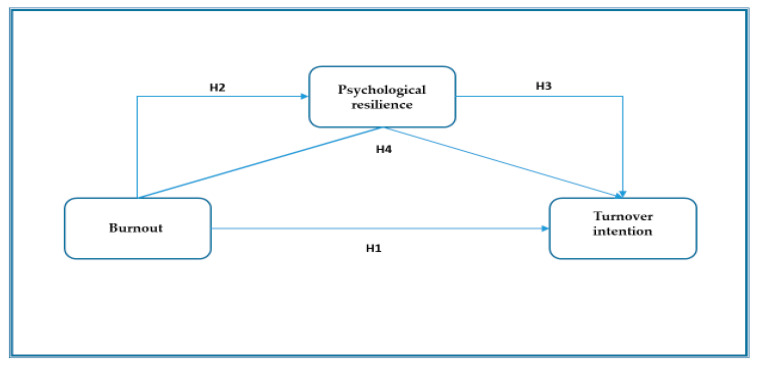
The research model and hypotheses.

**Table 1 behavsci-14-00782-t001:** Discriminant Validity Fornell–Larcker Criterion.

Variables	Turnover Intention	Burnout	Psychological Resilience
Turnover intention	0.818		
Burnout	0.485	0.672	
Psychological resilience	0.086	0.397	0.730

**Table 2 behavsci-14-00782-t002:** Construct reliability and validity.

Variables	CA	CR (rho_a)	CR (rho_c)	AVE
Turnover intention	0.928	0.937	0.923	0.670
Burnout	0.924	0.928	0.924	0.552
Psychological resilience	0.953	0.961	0.944	0.533

Notes: CA = Cronbach’s alpha; CR = composite reliability; AVE = average variance extracted.

**Table 3 behavsci-14-00782-t003:** Mean, standard deviation, and correlation values.

	Mean	Standard Deviation	1	2	3
1. Turnover intention	2.81	0.98	-		
2. Psychological resilience	3.36	0.77	0.04	-	
3. Burnout	3.38	0.78	0.44 **	0.26 **	-

Note: ** *p* < 0.01 (two-tailed).

**Table 4 behavsci-14-00782-t004:** Path analysis results.

	β	R^2^	Standard Deviation	T Statistics	*p* Values
BRN -> TI	0.481	0.494	0.071	6.798	0.000
BRN -> PR	0.375	0.381	0.089	4.209	0.000
PR -> TI	−0.117	−0.118	0.084	1.395	0.163
BRN -> PR -> TI	−0.044	−0.046	0.036	1.223	0.221

Notes: Bootstrap sampling size 5000; BRN = burnout; TI = turnover intention; PR = psychological resilience.

**Table 5 behavsci-14-00782-t005:** Model fit.

	Saturated Model	Estimated Model
SRMR	0.01	0.01
d_ULS	11.795	11.802
d_G	2.987	2.974
Chi-square	5.827	5.751
NFI	0.932	0.942

Notes: SRMR = standardized root mean squared residual; NFI = normed fit index; d_ULS = the squared Euclidean distance; d_G = the geodesic distance.

## Data Availability

Data are unavailable due to privacy or ethical restrictions; they can be shared on demand.

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
