# Peer review of "The Impact of Intensive Care Unit Nurses’ Burnout Levels on Turnover Intention and the Mediating Role of Psychological Resilience"

_behavsci, 2024, doi:10.3390/bs14090782_

Round 1
Reviewer 1 Report
Comments and Suggestions for Authors
Comments on the Quality of English LanguageAuthor Response
Thanks to the reviewer for his/her support to improve the quality of the manuscript. The corrections are in the atteched doc. The changes in the manuscript is written is green font. In addition, extensive english editing is done by 3 different professionals.

Reviewer 2 Report
Comments and Suggestions for Authors
Aspects that could be improved:
The statistical analysis is appropriate for the study's objectives. However, it could be enriched with a more in-depth practical interpretation of the statistical results.
The version of SPSS used should be mentioned.
The Discussion could be enriched by exploring other variables that might mediate the relationship between burnout and turnover intention, such as age, gender, tenure, or by offering these suggestions for future research. Include a more in-depth critical analysis of the study's limitations and how they may have influenced the results, as well as possible biases. Nothing is mentioned about Hypothesis 4.
The Conclusion could include more specific recommendations for healthcare managers and policymakers on how to address burnout among ICU nurses.
The formatting of the article should be reviewed, namely:
- Line 74: remove the following period: Greenhaus et al.'s [23]. work-life balance theory.
- Line 86: remove the periods and the parenthesis: framework of Maslach and Leiter's [22]. work-life model and Greenhaus et al.'s [26].)
- Lines 119, 183, 184, 186, 199, 280, 296, 339: there is an extra space
- Line 145: remove the word "second": The study therefore proposes the second third hypothesis:
- Line 153: separate turnover(64): between nurses' well-being, resilience, and turnover[64].
- Line 246: the beginning of the sentence is missing
- Line 329: remove one of the periods: turnover intention..
Review the writing style of the text; some sentences need clarification to avoid confusion in understanding the idea.
Comments on the Quality of English LanguageReview the writing style of the text; some sentences need clarification to avoid confusion in understanding the idea.
Author Response
Thanks to the reviewer for his/her support to improve the quality of the manuscript. The corrections are in the attached doc. The changes in the manuscript is written is green font. In addition, extensive english editing is done by 3 different professionals.

Reviewer 3 Report
Comments and Suggestions for Authors
The authors of this manuscript evaluated the burnout levels of intensive care unit nurses. The manuscript is interesting but it lacks some critical aspects. The following are some of my concerns:
1. Abstract should be re-written highlighting the results. Currently, the details on how the results are evaluated are not present. Some numeric results can be highlighted. The conclusion in the abstract can be shortened.
2. What is the work experience of the nurses involved in this study?
3. How was this survey conducted? Is it through google forms or ?
4. Details of the questionnaire should be included.
5. The statistical analyses of emotional exhaustion, depersonalization, etc., should be reported.
6. More details such as socio-demographic information of the nurses who participate in this study could be included.
7. The results section seems to be insufficient. More statistical analyses are required. Refer https://www.sciencedirect.com/science/article/pii/S0964339722000490
8. Limitation of the work is not discussed.
9. Conclusions should be supported by results.
10. Many similar works are available, highlighting how this study is different from others and the need for this study should be discussed.
Comments on the Quality of English Language
Minor editing of the English language required
Author Response
Thanks to the reviewer for his/her support to improve the quality of the manuscript. The corrections are in the attached doc. The changes in the manuscript is written is green font. In addition, extensive english editing is done by 3 different proffessionals.

Round 2
Reviewer 1 Report
Comments and Suggestions for Authors
Dear authors, thank you for including my remarks. I still think the content of the manuscipt is really important. However, I am still concerned about the method: using MBI and psychological resilience scale as one dimensional scale, also reliability is sufficient to use the original scales.
1. Burnout is not a “new” concept in nursing. Maslach and Leiter have done research in this areas for at least 35 years.
2. Line 72-76: I still think this argument is not necessary for the argumentation of the paper. I would remove it
3. Sample: “The final size of the sample was determined as 212 individuals with a 95% trust interval and a 5% error margin [76]” and “ After excluding the missing participants (respondents who left the questionnaires blank or significantly incomplete), 228 questionnaires were found suitable for analysis” à I don’t really understand this two sentences. What is the final sample size?
4. Line 162: “missing participants” is not the correct term. The participants are not missing. Their questionnaires had missing answers. Better use: “missing data” or something like this.
5. Method: I am still concerned about the one scale solution for the MBI and the psychological resilience scale. The reliability mentioned in the method is sufficient to use the scales separately. Both instruments have been established and proven to have a multidimensional structure, using a one-dimensional scale takes away variability and is methodical not necessary. I also believe using the resilience scale as one construct might be the reason why resilience seems to play no role in the results. Please reconsider using the one-dimension approach.
6. Line 288: which similar studies? No references
7. Line 359-361: Unfortunately, I don’t understand the meaning of this sentence. What are “simple reasons”?
8. 368-371: please rephrase and use references - “superhuman”?
9. Line 396-401: Why was this section added?
10. Discussion in general: please add references to your arguments. I found some arguments without references or only simple statements with “some studies” or “literature” without citations.
Comments on the Quality of English LanguageNewly added sections need to be checked
Author Response
Thanks to the reviewer for his/her efforts to improve the quality of the manuscirpt. The corrections are listed in th attaeched document. The revisions are highlighted as green font in the manuscirpt.

Reviewer 3 Report
Comments and Suggestions for Authors
All of my concerns are adequately addressed. Just a few suggestions.
Instead of highlighting how this manuscript is different from similar studies in the conclusion section, it is better to highlight it in the introduction section.
The conclusion section should not have citations.
Author Response
Thanks to the reviewer for his/her efforts to improve the quality of the manuscript. The revisions are highlilghted as green font in the manuscript. Please see the corrections below.
1. Instead of highlighting how this manuscript is different from similar studies in the conclusion section, it is better to highlight it in the introduction section.
Since the other reviewers requested the difference of the study to be added into the conclusion section, this addition was made to the conclusion section. However, in order to meet the reviewer's valuable suggestion, a sentence has also been added to the introduction.
2..The conclusion section should not have citations.
The suggested corrections have been made.
Round 3
Reviewer 1 Report
Comments and Suggestions for Authors
While I believe the manuscript benefited strongly from the improvements made. My main concern remains with the one-scale options, as I truely believe it takes away variability of the scales and inhibits or masks specific results otherwise found. More specific, questions that I do believe could be answered otherwise would be: Does emotional exhaustion or depersonalization lead to higher turnover intention? Does self-efficacy moderate the relationship between personal accomplishment and turnover intentions but not emotional exhaustion and turnover intentions? With this more specific questions and answer, I believe more specific conclusions for interventions in the nursing workfield can be drawn.
However, as the authors did not change the results and explained their methological reasoning, I will not point this out anymore, if the editor approves of this procedure.
However, I would recommend to state this reasoning not only in the letter to the reviewer but with the statistical data from the factor analyses (inkl. SEM) in the paper.
Comments on the Quality of English LanguageMinor edits that can be completed by the journal are needed.
Author Response
While I believe the manuscript benefited strongly from the improvements made. My main concern remains with the one-scale options, as I truely believe it takes away variability of the scales and inhibits or masks specific results otherwise found. More specific, questions that I do believe could be answered otherwise would be: Does emotional exhaustion or depersonalization lead to higher turnover intention? Does self-efficacy moderate the relationship between personal accomplishment and turnover intentions but not emotional exhaustion and turnover intentions? With this more specific questions and answer, I believe more specific conclusions for interventions in the nursing workfield can be drawn.
However, I would recommend to state this reasoning not only in the letter to the reviewer but with the statistical data from the factor analyses (inkl. SEM) in the paper.
Answer :
Thank you for your efforts to improve the quality of the manuscript.
We certainly agree with the reviewer’s valuable opinions. To meet the concern with the one-scale option, we have added new paragraphs into methods section in red font which gives similar one dimension aproaches in the literature wwith 13 new citations.
As to the questions, to include asnwers to these questions, the study must be renewed, which is not possibe at the moment. We will take them into consideration in the future studies.